# Long-term spatial dynamics of jaguars in a high-density population

**Bart J. Harmsen** *, **Rebecca J. Foster**

Panthera, New York, New York, United States of America

* bharmsen68@gmail.com

## Abstract

We assessed the socio-spatial dynamics of a jaguar population over 15 years using camera-trap data from Belize. Using ~4,000 independent detections of male jaguars, we documented and quantified range shifts, overlap, and interactions between males. Additionally, ~700 independent detections of females allowed us to investigate interactions between the sexes. Using the distance between activity centres, we assessed the variation in space use within and between males. Male ranges were not stable: activity centres shifted from one year to the next, with a mean maximum distance moved of 8 km per individual between any two years (SD = ± 6 km, n = 371). Overall, we found no evidence of exclusive territoriality. Male jaguars overlapped extensively with one another: males shared, on average, half of their detected range with at least one other male, per year (Volume of Intersect, kernel overlap); while their activity centres lay within 2 km of at least one other male. Close encounters (two individuals at the same location within 24h) were most common between males whose activity centres were ≤ 4 km apart. We found that close encounters between prime males (3-7y) and old males (≥ 8y) occurred less frequently than expected, suggesting avoidance of stronger competitors by the physically weaker, older age class. Notably, old males also had fewer close encounters than expected with females; potentially having reduced access to females because they were avoiding areas frequented by prime males. However, we found no clear evidence of males monopolising females, as females tended to associate with more than one male during the short time windows when they were detected on trails. Overall these results are consistent with a system of scramble competition among males in which overlapping and shifting ranges result from searching for receptive females that are distributed sparsely and unevenly in space and time.

## Introduction

Conservation concern for large carnivores is widely acknowledged [1]. Because social factors play an important role in regulating carnivore population dynamics

**Data availability statement:** All relevant data are within the paper and its Supporting Information files.

**Funding:** The author(s) received no specific funding for this work.

**Competing interests:** The authors have declared that no competing interests exist.

(e.g., [2])an understanding of their social organisation can inform their management and conservation (e.g., [3]). This requires concurrent data on multiple individuals, collected across temporal scales: at a fine-scale to detect evidence of social interaction (avoidance, attraction, tolerance); and over the long-term to detect territoriality and assess range stability. Because felids are generally solitary and highly cryptic, such data are difficult to collect, thus their social systems are poorly understood [4,5]. Inferences about felid social systems are based on tracking data, direct observation, and studies of marking behaviour (e.g., [6–9]). Historically, these studies have been limited to a handful of species, populations and environmental conditions, focusing on gregarious and highly visible species (e.g., [4]). Although advances in GPS collar technology have facilitated rich movement data (e.g., pumas, *Puma concolor* [10,11]), inferences about social organisation are often constrained by the low number of individuals tracked simultaneously and the duration of monitoring, particularly for more elusive species (e.g., jaguars, *Panthera onca*, [12,13]). Here, we demonstrate the utility of camera traps for studying the long-term space use patterns and dynamic interactions of cryptic felids.

Within the context of social organisation, carnivore space use patterns reveal the differing reproductive strategies of males and females. For most solitary carnivores, including felids, females provide the parental care, and reproductive success depends on the number of off-spring raised to dispersal age [14,15]. In contrast, male reproductive success, if providing no parental care, depends on the number of successful fertilisations achieved in their lifetime (e.g., [16]). Socioecological models therefore predict that the female range will encompass sufficient prey and refuges for herself and dependents, within the constraints of pregnancy, lactation, and the limited mobility of her cubs (e.g., [17,18]). In contrast, males are expected to disperse further than their female siblings, maximising access to unrelated fertile females while maintaining enough food for survival [19–22]). Within this general model, reproductive strategies may show a high degree of plasticity, reflected in different patterns of space use. E.g. female space use may change when dependents have dispersed or when the female is in oestrus and actively seeking mates (e.g., [20]); and male space use varies from roaming widely, scrambling to mate with different females throughout their lifetime; to defending a territory, monopolizing one or several females with which they repeatedly sire litters [7,23,24].

The intense competition between males for reproductive advantage manifests as aggression towards rivals for mating access (e.g., [16,19]). The physical competitive ability of males varies through time, increasing with maturity, and deteriorating as they senesce (e.g., [25]). Because prime males have physical competitive advantage over younger and older males, we expect male strategy, and hence space use, to vary with age and experience. Under certain conditions, they may also form male-male coalitions [17]. The intense competition between males for reproductive advantage is also expressed as sexual coercion and aggression towards females ('sexual disturbance' sensu [26]) and infanticide of cubs fathered by competitors [27,28]. Consequently, female space use is influenced by strategies to minimise the direct physical harm and indirect costs of sexual disturbance and infanticide, such as

avoiding or hiding from males when raising cubs, forming female coalitions, or associating with multiple males to confuse paternity (e.g., [7,28,29]).

By monitoring multiple individuals simultaneously throughout their lives, long-term data can address questions about socio-spatial dynamics. In open habitats, felid social systems have been elucidated by direct observation of habituated individuals (e.g., leopards, *Panthera pardus*, [30]). In closed habitats, such as the dense jungles of the neotropics, direct observation of felids is not possible, and attempts to study their socio-spatial patterns have traditionally relied on telemetry (e.g., jaguars [31]; ocelots, [32]). However, few, if any, studies track individuals throughout their entire lifetime, sample sizes are generally low, and the study sites are often restricted to the more open habitats where live trapping is logistically feasible. E.g., of the 117 jaguars that have been tracked using GPS collars from 20 independent study sites [12], approximately three-quarters (72%; 84/117) were monitored for less than one year, and only four individuals were monitored for at least 3 years. Only five studies have tracked 10 or more jaguars, all of which were in Brazil; and 80% in the Pantanal, a relatively open wetland ecosystem that contrasts with the tropical moist forest system comprising the majority of the jaguars' geographic range. Camera-trap data have the potential to provide a non-invasive and cheaper addition to telemetry for studying space use and interactions under these conditions. Although camera trapping only provides a crude representation of spatial range, limited by the extent of the camera grid, this is compensated for by the continual, simultaneous long-term monitoring effort of passive detectors at multiple fixed sites.

In this study, we use camera-trap data spanning 15 years, to understand the social system of a stable, high-density population of jaguars inhabiting the moist tropical forest of the Cockscomb Basin Wildlife Sanctuary in Belize [33–35]. Turnover is low and tenure of both sexes in the study area is long, with apparent adult survival rates of 0.78 [34] We monitored the population over multiple cohorts, following some individuals through their entire life. This allowed us to investigate flux in tenure and associations between individuals, and thereby draw inferences about the social system and develop hypotheses about the mechanisms governing the observed space use. If males reproductive success is maximised by defending female ranges against other males, then we expect males will be territorial. However, if the cost of territorial defence exceeds the gain, then male home ranges will overlap. On this basis, [36] argued that if female reproduction is synchronous (receptive females are clumped in time), then the male spacing system will be territorial, with either a monogamous or polygamous mating system, depending on whether females are widely dispersed or overlap. In contrast, if female reproduction is asynchronous (receptive females are dispersed in time), then the males will have overlapping ranges, and the mating system will be promiscuous. Female jaguars are polyoestrous, and in tropical areas such as Belize, where there is no marked seasonality in day length, reproduction can occur at any time of the year [37–39]. Therefore, assuming asynchronous reproductive cycles and an even distribution of receptive and non-receptive females in space and time, we expected an overlapping system of males, and a promiscuous mating system. Early work in the study area suggested that males do not defend exclusive territories, rather their ranges overlap and shift through time [31,40]; suggesting that they do not defend female ranges. However, these studies were relatively short (telemetry of five males for up to nine months; and camera trapping of 23 males for up to 3 years) providing no evidence that individuals follow such strategies throughout their lifetimes.

Our goal was to describe the long-term space use and dynamic interactions of male jaguars to draw inferences about the social system of a high-density jaguar population. Because the population is stable and the prey base not considered limiting [34,41], we assume that flux in patterns of space use reflect movement associated with reproductive strategy rather than a response to density-dependent drivers such as varying availability of prey. We investigated (1) the stability of individual ranges over multiple survey years, (2) the degree of overlap between individuals and the number of individuals they overlap with (crowdedness) within areas of overlap, (3) evidence for avoidance, attraction and/or tolerance between individuals, (4) the demographics (age and sex) of individuals involved in encounters. If prime males are territorial, excluding rivals, monopolising females, and defending against infanticide, we predicted that we would detect stable ranges over multiple survey years for these males. Their ranges would be characterised by low male-male overlap and low

crowdedness but with evidence of encounters between male neighbours at range boundaries, and movement of neighbours into unoccupied areas when residents vacate the area. In contrast, under conditions of scramble competition, in which all males maximise access to receptive females and there is no year-round defence of female ranges, we predicted shifting ranges independent of the movements of other males. In this scenario, males ranges would be characterised by high male-male overlap and crowdedness, with males of any age encountering other males; and single females encountering multiple males during oestrus. On the basis of these metrics, we assess space use through time within males and between males, and their encounters with females.

## Methods

### Study area

The study was conducted in the Cockscomb Basin Wildlife Sanctuary in Belize (here-on, CBWS or sanctuary), 490 km$^2$ of moist broad-leaved tropical forest. The sanctuary is part of the Maya Mountain Massif Jaguar Conservation Unit (MMM JCU), consisting of 5,000 km$^2$ of forest without human habitation. The CBWS was selectively logged until 1981, officially protected in 1986, and is now a mosaic of advanced secondary forest in several stages of succession. Many of the old logging roads in the eastern basin are maintained as tourist trails or patrol routes (Fig 1), providing easy, and presumably preferred, travel routes for jaguars to move through the dense secondary vegetation [31,34,42]. Relative to other tropical moist broad-leaved sites, the Cockscomb Basin supports a high density of jaguars, with between 20–30 individual animals detected using the study area annually, and density estimates ranging from 2 to 8 jaguars/100 km$^2$ [34,43].

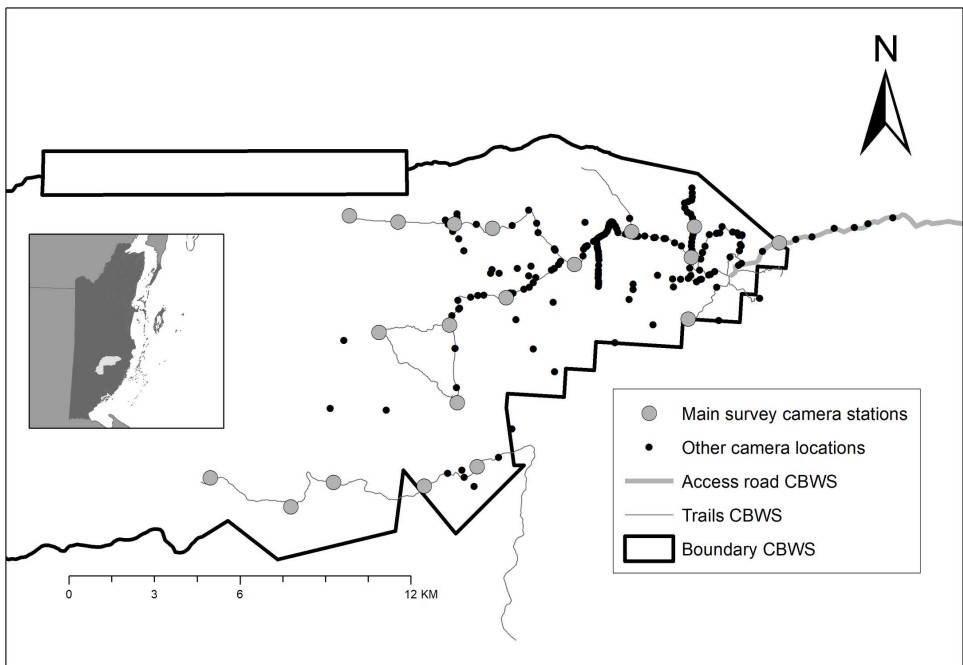

**Fig 1. Study area in the Cockscomb Basin Wildlife Sanctuary (CBWS), Belize (see inset map), showing the trail system and distribution of camera stations from 2002 to 2017 for 20 fixed locations used for annual main survey (large grey circle) and supplementary camera locations used for small-scale surveys (small black circles).**

## Camera trapping

We used 20 fixed locations, with two camera traps per station, to survey during the February-June dry season from 2002 to 2008, and from 2011 to 2017, covering an area of ~120 km$^2$ (Fig 1, [34]. This spans a male jaguar home range based on GPS collar data from two adult males using the study area during the study period (annual range: 150 and 114 km$^2$, mean 3-month (quarterly) range: 100 and 85 km$^2$; [35]). Neighbouring stations were separated by 1.07 to 3.05 km (mean = 2.02 km); the furthest distance between any two stations was 21.6 km. Surveys lasted 59–98 days, except for the 2013 survey which ran for one year (365 days, N = 1) to explore seasonal variation in detection probability [35]. The study design identified optimal locations with high capture probability and these camera locations were maintained over the years [33–35,43]. We supplemented our 20-location dry season camera survey (from here on 'main survey') with additional data from small-scale camera- and video-trap surveys deployed inside and outside of the sanctuary, resulting in a database of 210 monitored locations [8,34,44,45]. From 2002 to 2008, we used traditional film camera traps (CamTrakker, Cuddeback and DeerCam) with an enforced 3-minute delay between exposures to prevent excessive photos of herding species such as peccaries (*Tayassu* sp.). This might have negatively biased our ability to detect sequences of multiple jaguars following one another over extremely short periods. From 2011 onwards, we used digital camera traps (Pantheracam; Panthera corp, New York, NY) with a minimum delay of 8 seconds between successive photo triggers. Data from this period indicated no sequential captures of jaguars within the 3-minute period, indicating that bias using the older type cameras, with 3 minute forced delay, was likely low. We checked and changed cameras, batteries, film and SD cards as required every 2–4 weeks depending on the location and the camera model. Every photograph was stamped with the time and date. We identified individual jaguars based on their unique spot patterns, and assigned sex based on the presence or absence of testicles, following [46]. The number of jaguars that we were unable to identify up to individual level was very low (mean 1.5% of total detections per year and maximum of 5.9% of detections per year, n = 14). We are therefore comfortable with the assumption that we were able to identify almost all detections to individual level.

## Spatial overlap

Camera traps sample the movement of individuals from fixed locations. Therefore, estimates of home range size and shape from camera detections are constrained by the size and layout of the camera grid. However, if the camera grid is dense, and individual detection probabilities are high, it is reasonable to assume that the spatial pattern of detections reflects crude range use within the study area. From the main survey, we used the detections of male individuals, defined as a single detection per camera location within 24 hours, to estimate activity centres (centroids) as proxies of range use. This allowed us to assess consistency or variation in range use and overlap within and between individuals through time. We acknowledge two factors which may bias our interpretation of range use: (1) distance between activity centres may be underestimated at the edge of the study area for individuals whose range extends beyond the camera grid, and (2) as the main survey cameras were all trail-based, our study of range use is limited to movement along trails, therefore any assessment of overlap refers explicitly to use of trails rather than the forest matrix per se.

We used the distance between activity centres as a proxy for range overlap. We calculated the activity centre of each male per year ('annual activity centre'), as the mean of the x-coordinates and y-coordinates of all of its independent detections, including repeated independent detections at the same location. We limited our analysis to individuals with at least seven detections per year so as to ensure sufficient detections to calculate their activity centres. We used the Geosphere package in program R [47] to calculate distances between all possible pairs of activity centres (within individuals between years, and between individuals within years). We tested whether the distances between activity centres were a valid proxy for range overlap by testing for a correlation with estimates of overlap derived using the volume of intersect (VI) between yearly home-range kernels (a more data-hungry method, therefore restricted to the fewer individuals with more detections; see Supplementary Methods (S1_File)) and found a strong and significant correlation between the two methods (r = −0.90, p < 0.01, n = 152).

*Shifts in male range use* – For each male with >1 annual activity centre, we assessed activity centre stability by calculating the distance between annual activity centres. For the subset of males with at least five consecutive years of activity centres, we also calculated the maximum distance, mean distance, and trimmed mean (i.e., excluding the maximum value) distance moved between consecutive years.

*Overlap and crowding between males* – We assessed the overlap per individual as the minimum distance from another male's activity centre. We defined neighbours as males with activity centres separated by ≤ 2 km; equivalent to 75% overlap, assuming 100 km$^2$ circular home ranges (see Supplementary Methods (S1_File)). We assessed crowding for each individual by summing its number of neighbours and averaging over the number of years it was detected. Together, these two measures allowed us to assess the extent of overlap and crowding within the Cockscomb jaguar population, and variability between individuals and through time. We assessed the stability of neighbouring ranges across years by summing the number of years that pairs of individuals with at least five years of data, maintained activity centres that were ≤ 2 km apart.

Shifts in activity centres and neighbour identities through time, and high overlap and crowdedness, would provide evidence for instability in range use, suggesting that all males attempt to maximise access to receptive females and no territorial defence of female ranges. In contrast, stable ranges over multiple survey years, with individuals interacting with the same neighbour for extended periods, low male-male overlap and low crowdedness, would suggest that males are territorial.

## Male-male interactions

*Evidence for avoidance* – We sought evidence for intraspecific avoidance versus indifference or attraction by assessing the chronological order in which males passed camera locations. Using consecutive detections of male jaguars at the same location, we assessed the pattern at which the various individuals passed these locations. Clustered detections of the same individual at the same location would indicate avoidance, meaning different individuals took turns in using the location. On the other hand, consecutive detections of different individuals would indicate indifference or attraction, with individuals intermingling at the same location. For each camera location, we recorded every male detection in chronological order, noting the individual's identity. Using this time series of identities, we summed the number of consecutive detections of the same individual (Jaguar A followed by Jaguar A, 'same') versus the number of consecutive detections of different individuals (Jaguar A followed by Jaguar B, 'diff') at each camera location. We compared the number of 'same' sequences versus 'diff' sequences, per camera location, using the paired Wilcoxon test in R [48]. A higher number of 'same' sequences per camera location would suggest that individuals avoid one another (camera locations are dominated by a single individuals) while a higher number of 'diff' sequences would provide evidence for use of same location by multiple individuals and thus the possibility for interaction (direct or indirect) between conspecifics.

*Evidence for attraction* – To assess whether male jaguars using the same location were indifferent or attracted to one another, we recorded the lengths of the 'diff' sequences and categorised them into 1-day, 2-day, 3-day… x-day intervals (bins). For each individual associated with at least 15 'diff' sequences pooled across all camera locations, we calculated the proportion of their 'diff' sequences in each bin, and assessed the distribution. A right-skew (short time intervals between consecutive detections of different individuals) would provide evidence for males following/investigating each other, or being attracted by the same resource. A normal distribution would suggest that males are moving randomly with respect to the location of conspecifics (indifference). A left-skew would suggest that males are temporally avoiding one another. We repeated the analysis using individuals with at least 6 'diff' sequences (more individuals, fewer interactions per individual), and using individuals with at least 30 'diff' sequences (fewer individuals, but with more interactions per individual), and found no difference in the resulting distribution. We tested for normality of the distribution using the Shapiro-Wilk test [48].

*Evidence for territoriality* – We sought evidence for territoriality by investigating whether the likelihood of males encountering one another ('close encounters') was associated with (1) their age, and (2) the distance between their activity centres. We defined 'close' encounters as 'diff' sequences with time intervals ≤ 24 hours (i.e., consecutive detections of two different males within 24 hours of each other at the same location). Twenty four hours is the frequently cited cut-off point in camera trap density literature for independence of jaguar detections (e.g., [34]).

*[1] Age* – Age was calculated as number of years since first detection + 2, assuming that individuals were at least 2 years old when first detected as an adult. We recognise the bias of underestimating age if certain individuals were first detected when they were older than 2 years. However, our long-term data show that first detection is mostly at a young age, supported evidence of young immigrants from neighbouring study areas establishing ranges at a young age in our study area (BHarmsen Unpubl. data). We always validated age class of first detection based on photographs, subjectively assigning age class based on head width, muscularity and testicular size. If males proactively defend areas then the probability of close encounters between prime males would be higher than the random expectation (attraction at sites of area disputes); and close encounters between prime males and the older or younger age classes would be equal or less than random, reflecting tolerance or avoidance respectively. Conversely, under conditions of scramble competition, we expect that males of any age would encounter one another in accordance with their detection probability along the trail system. For every close encounter we categorised adult males as young (2y), prime (3 to 7y) and old (≥ 8y) following [34], and recorded the age class of the dyad: young with young, young with prime, young with old, prime with prime, prime with old, or old with old. Using the probability of detection for each age class (*number of detections in age class x/total number of detections*), we calculated the expected detection probability of each age class dyad. We used a chi-squared goodness of fit test [48] to test whether the observed distribution of age class dyads differed significantly from the expected distribution. We repeated the analysis for the subset of close encounters that occurred within ≤ 15 minutes, from here-on 'true' encounters.

*[2] Distance between activity centres* – We investigated how the frequency of close encounters varied with the distance between activity centres. If males defend exclusive territories then we expect that the majority of close encounters would occur with neighbours at the territory boundaries, giving rise to a narrow, bell-shaped distribution with short tails, with few encounters at larger distances. Under this scenario, we would expect a peak in close encounters at a distance that reflects the average home range radius. Assuming a circular range of 100km², average male home range radius in the study area is approximately 5.5 km [35]. Therefore, under conditions of exclusive territoriality we would expect a peak in close encounters at distances of approximately 10–11 km between activity centres. If males overlap widely and do not defend exclusive territories, then we expect the frequency of close encounters to be inversely proportional to the distance between activity centres, with close encounters occurring more frequently between individuals whose activity centres are closer in space, giving rise to a right-skewed frequency distribution with a long tail. We tested for normality of the distribution using the Shapiro-Wilk test [48].

## Male-female interactions

Detections of females were too few to investigate range use and interactions across and within years; however, we assessed whether close encounters between males and females were associated with the ranging behaviour and/or age class of males. We hypothesised that mobile males in prime age would encounter more females and more frequently, than would older, less mobile males. We considered repeat encounters of the same male-female pair at the same station independent if they were separated by ≥ 1 week, based on the duration of female receptivity [38]; therefore, we excluded from the analyses any repeat detections that occurred at the same station during the same week. We assumed that detections between young males would reflect mother-offspring relationships, therefore we excluded encounters involving males during their first year of detection (assumed to be ≤ 2 years old).

For every independent close encounter with a female, we recorded the age class of the male as either prime or old and used a goodness of fit test to test whether the observed distribution differed significantly from the expected distribution

based on the probability of detection of males in each age class. We repeated the analysis for the subset of true encounters.

Using the subset of males with at least five consecutive years of capture history, we tested for correlation between the number of close encounters with females and (a) the mean distance a male moved between years, and (b) the maximum distance a male moved between years across its lifetime, using Pearson correlation [48]. To check for long-term male-female associations, we recorded the number of years over which these males repeatedly encountered the same female(s). We also sought evidence of males monopolising females. For each female, we recorded the number of independent close-encounters with males and the number of different males encountered, (1) over their monitored lifetime and (2) within a fixed period of 20 days from being detected with a male, chosen to reflect the maximum possible mating period of jaguars, soliciting interest from males [49–52].

## Results

From 2002 until 2017, we detected 136 jaguars (46 females, 71 males, and 19 of unknown sex); of these, 38 males were each detected at least 7 times within a given year (mean 19.7 detections/year, range 7–98, n = 38). For these males, we estimated their annual activity centres (1–11 annual activity centres per individual; mean = 3.1, SD = 2.3, n = 118) and the distance between every possible combination of pairs (within and between individuals). The number of jaguars that we were unable to identify up to individual level was very low (mean 1.5% of total detections per year and maximum of 5.9% of detections per year). We are therefore comfortable with the assumption that we were able to identify almost all detections to individual level.

### Spatial overlap

*Shifts in male range use* – We had sufficient multi-year data from 26 male jaguars for comparison of activity centres within individuals between years (2–11 years per individual). On average, an individual's annual activity centres moved by 3.9 km between any pair of years (SD = 2.2 km, n = 371 pairwise comparisons), with a mean maximum distance between any two years of 7.7 km per individual (SD = 5.5 km, maximum = 21.6 km, which was the widest extent of the camera grid, n = 371 pairwise comparisons), meaning that the focal location of male jaguars' activity shifted through time. For the 11 male jaguars with at least 5 consecutive years of activity centres, individuals' annual activity centres shifted to varying degrees through time. While some remained in the same local area, having a small spatial extent over their lifetime, others drifted gradually through time, or made a single large move (spatial extent of individuals' annual activity centres: mean = 9.4 km, SD = 3.9, range = 4 to 17.1; S1 Fig). As the largest moves within this dataset were found in the time frame between 2008 and 2011, across the missing survey years, moves here were likely more gradual between these points, as measured across a longer time frame. When removing these and only assessing consecutive years, the maximum move between years ranged from 3.5 to 8.4 km across individuals (mean = 5.9, SD = 1.5, n = 11 jaguars), which is still considerable, with nine individuals shifting their activity centre by more than 5 km from one year to the next. Maximum moves in activity centre occurred at different periods during tenure, with two individuals moving maximally at the start of their tenure, two at the end, and the remainder during the middle (S1 Fig). Excluding each individual's maximum move between consecutive years, the mean distance moved from one year to the next ranged from 0.5 to 2.9 km across individuals (n = 11 jaguars), with seven individuals shifting their activity centres, on average, less than 2 km each year, suggesting periods of relative stability in range use, intersected with larger moves. Overall, we found evidence of male ranges shifting across the years.

*Overlap and crowding between males* – We had sufficient data for 38 male jaguars to calculate the distance between pairs of conspecific activity centres within years. On average, the mean minimum distance detected between the activity centre of any two males per year was 1.8 km (SD = 1.4, range 0 to 8.6 km, n = 118). On average, males were within 2 km of at least one other male (i.e., had at least one neighbour) each year (level of crowding: mean number of

neighbours = 0.93, SD = 0.77, maximum = 3). The highest level of crowding we recorded (3 neighbours within a year) was detected for two individuals, and we recorded 23 instances of 16 males each having two neighbours within in a given year, giving an overall clustered pattern of activity centres (S2 Fig).

Dyads in which males were neighbours (activity centres ≤ 2 km apart) had a low level of permanency through time, never exceeding 2 years and none of these two years concerned two consecutive years in a row (always staggered with other neighbour dyads in between). For the 22 males for which we could assess activity centres across multiple years (≥ 5 years, maximum 11 years), 59% (13/22) had the same neighbour for 2 years, 36% (8/22) had different neighbours each year, and one never had a neighbour; illustrating a high level of flux in range use and neighbours.

## Male-male interactions

*Evidence for avoidance* – Across 210 camera locations, we recorded 2,797 consecutive detections of different males ('diff' sequences) and 1,693 consecutive detections of the same males ('same' sequences). Regardless of location, consecutive detections of different males were more common than consecutive detections of the same males (Wilcoxon signed-rank test $V = 15854$, $p < 0.0001$, median "diff" = 4 sequences per camera location (3rd Q = 10), median "same" = 2 per camera trap (3rd Q = 6)), suggesting that males were monitoring one another or attracted to the same shared resource.

*Evidence for attraction* – For the 37 male jaguars with at least 15 'diff' sequences, the distribution of "diff" sequences was right-skewed towards short time intervals (Shapiro-Wilk normality test $W = 0.73$, $p < 0.001$), with, on average, 17% (SD = 0.08) detected at the same location within 24 hours of each other ('close encounters'; Fig 2). During the 15-year study period, we detected, on average, 2.8 close encounters per male-male dyad (SD = 3.3, n = 204), with a maximum of 18 close encounters. The longest period across which we detected close encounters for the same male-male dyad was 5 years (mean = 1.4, SD = 0.8 years per male-male dyad, n = 204).

*Evidence for territoriality (1) Age* – Of the 6,986 male jaguar detections, 9% were young males (2y), 65% were prime males (3-7y) and 26% were old males (8 + y). Overall, we detected 385 close encounters between male jaguars, of which 32 were considered 'true' encounters (within 15 minutes of each other). We found no evidence of territorial behaviour among prime males. The number of close encounters exclusively between prime males, exclusively between old males, and between young and prime males were as expected given their detection probabilities on the trail system (Fig 3). In contrast, the number of close encounters exclusively between young males, and between young and old males, were more than expected; while the number of close encounters between prime and old males was less than expected (Chi-Sq = 45.2, df = 5, p < 0.0001, n = 385 close encounters; Fig 3a). The number of true encounters was as expected for all age class combinations, with the exception of encounters between young and old males, which was more than expected (Chi-Sq = 28.6, df = 5, p < 0.0001, n = 32 true encounters; Fig 3b).

*Evidence for territoriality (2) Distance between activity centres* – We detected 237 close encounters between 37 males for whom we could assign activity centres. For these males, the majority of close encounters (158/237, 67%) occurred between males whose activity centres were ≤ 4 km apart; and two-thirds of those encounters (102/158, 65%) were between 'neighbouring' males (activity centres ≤ 2 km; Fig 4). Only four close encounters occurred between males whose activity centres were ≥ 10 km apart. The distribution deviated from normality based on the Shapiro-Wilk test (W = 0.83, p < 0.001).

Few individuals maintained consistent dyad pairs across the years, both in terms of spatial overlap and close encounters. In the Supplementary Case Studies (S2_File), we provide some specific examples of associations between pairs of males.

## Male-female interactions

A total of 25 adult males had 112 close encounters with females, of which 20 were true encounters. Old males had fewer close encounters (and true encounters) with females than expected; while prime males had as many as expected, given their detection probability on the trail system (close: Chi-Sq = 4.8, df = 1, p < 0.03, n = 112; true: Chi-Sq = 4.3, df = 1, p < 0.04, n = 20; Table 1).

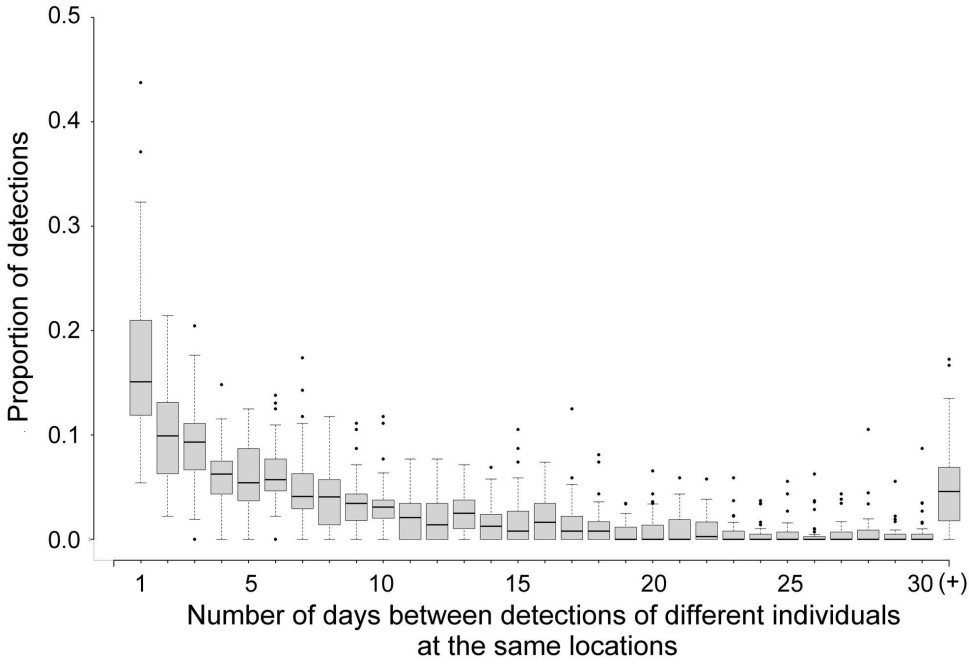

**Fig 2. Proportion of consecutive detections of individual X with any other individual within 1, 2, 3 … 30(+) days of each other; mean (horizontal bars) shown for 37 individuals, the means (black line in the grey box) of all bins sum to 1, the grey box represents 50% of all the data on either side of the mean with the whisker lines extending outside of the grey box, showing the maximum and minimum range, excluding outliers 1.5\* interquartile distance from the median. Outlier points outside of this range are indicated as points above and below the whisker lines.**

The subset of males detected for at least five consecutive years had close encounters with an average of three females across their lifetime (range 1–6 females, n = 11 males). Of these, five males (46%) had repeated encounters with the same female individuals, frequently across multiple years (Mean = 1.6 years, SD = 0.8, range 1–3).We found no relationship between either the maximum annual distance moved or total life time distance moved by males and the number of close encounters that they had with females, or the number of different females that they encountered (Pearson correlation, all tests r < 0.4, p > 0.2, n = 11 males).

Overall, we found no clear evidence of males monopolising females, as females tended to associate with more than one male during the short time windows when they were detected on trails, presumably in oestrus (pro-receptive or receptive). Over their monitored lifetimes, we detected 21 out of 30 females having multiple close encounters with males, averaging 5.4 encounters (range 2–23) with 3 males (range 1–8). For the remaining nine females, we documented just one lifetime close encounter each.

Additionally, we found 19 instances of 14 females having multiple close encounters with multiple males (max 2) within a 20-day period ('mating' period). On average, they had three close encounters (range 2–7) over seven days (range 1–20). We also found seven instances of five females having two close encounters with just one male each over an average period of 3 days (range 1–14).

## Discussion

To our knowledge, this is the first long-term study of felid social dynamics using camera trap data to follow entire cohorts throughout their lives. Our study demonstrates that long-term camera trap surveys can provide rich data on the socio-spatial dynamics of otherwise elusive felids. To our knowledge, this is the first such study to document and quantify the

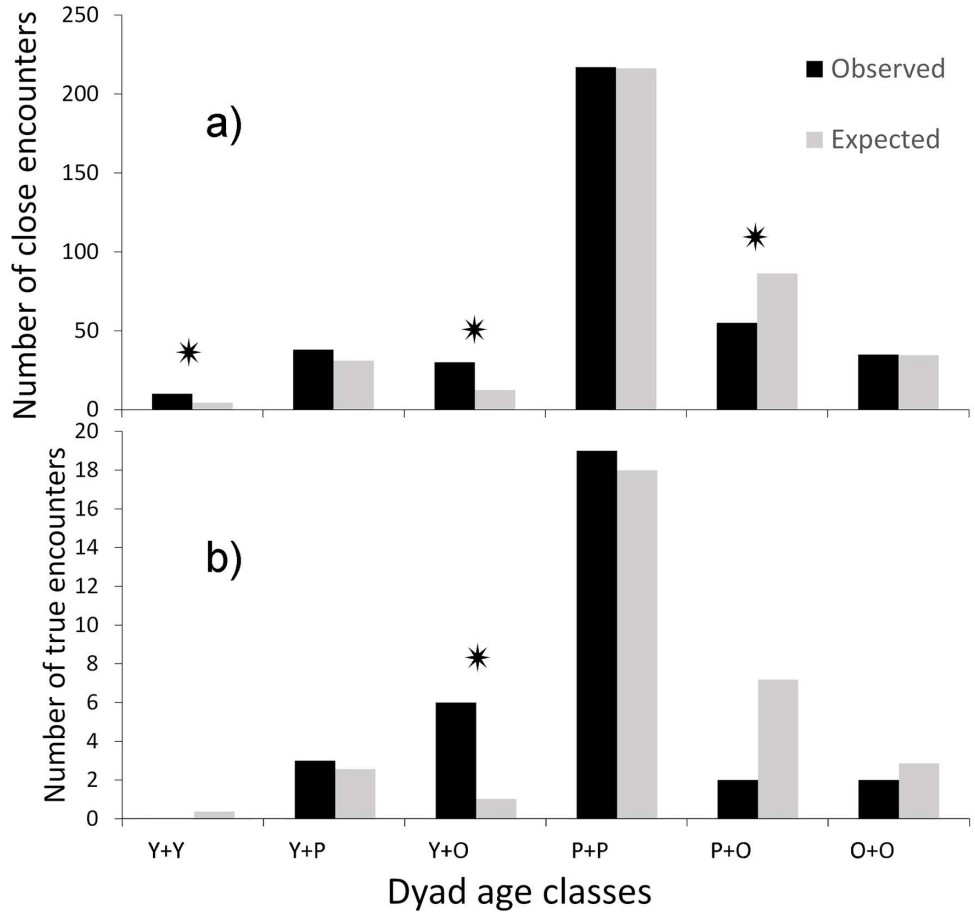

**Fig 3. Observed (black) and expected (grey) frequency of close encounters and true encounters a) close encounters (detections at the same location within 24h) between male jaguars, and b) true encounters (detections at the same location within 15 minutes) between male jaguars; by age class (Y=young, 2y; P=prime, 3-7y; and O=old, 8+y); * indicates that observed was significantly different than the expected at p<0.001.**

range shifts, overlap and interactions of jaguars over a 15-year period. We found that the focal location of male jaguars' activity shifted from one year to the next and male space use was characterised by high male-male overlap and crowdedness along trails. We found no evidence that males monopolised encounters with females, rather females encountered multiple males during the short periods that they were detected on trails. Such behaviour is expected in a system where males are scrambling to compete for receptive females that are sparsely and unevenly distributed in space and time [36,53].

We found no evidence of exclusive territoriality among male jaguars, and despite the crowdedness, there is no evidence of physical aggression in this system [34]. Under prey-rich conditions in a high-density population such as Cockscomb [34,41], defending exclusive territories may become too costly, resulting in risk aversion and a high level of tolerance (e.g., leopards, [30]). Indeed, males overlapped extensively in space, sharing at least half of their detected annual range with at least one other male. This high level of spatial overlap reflects the use of the trail system by multiple males. The trail system allows jaguars to more easily traverse the otherwise dense forest [45]. For felids, well-travelled trails also allow the efficient transfer of information via scent-marking [9]. In Cockscomb, jaguars mark along trails. Among the males this behaviour is associated with dominance and the presence of females, and their return rate increases in

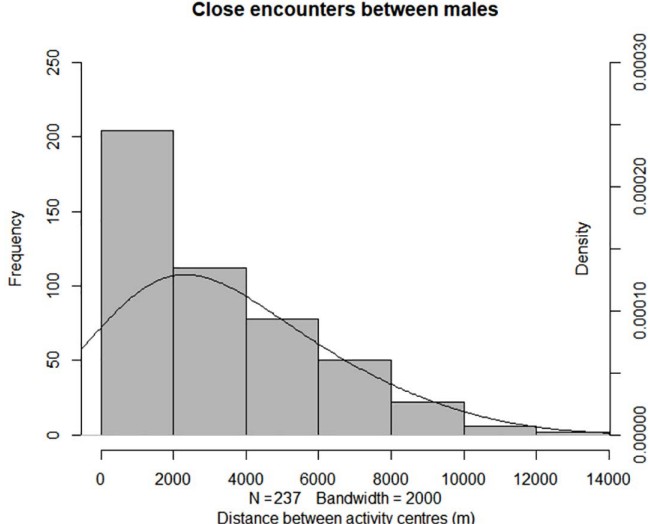

**Fig 4. Frequency of close encounters between 37 male jaguars according to the distance between their activity centres, n = 237 close encounters.**

**Table 1. Frequency of observed and expected encounters between male and female jaguars, according to male age class (prime, 3-7y; old 8 + y). a) Detected at the same location within 24h (close encounters, upper panel). b) Detected at the same location within 15 min (true encounters, lower panel).**

| a) Male age class | Observed | Expected | Chi-Sq contribution | Std resids |
|---|---|---|---|---|
| Prime | 91 | 80 | 1.50 | +1.23 |
| Old | 21 | 32 | 3.76 | −1.94 |
| **TOTAL** | **112** | | | |
| b) | | | | |
| Prime | 19 | 14 | 1.55 | +1.25 |
| Old | 1 | 6 | 3.76 | −1.97 |
| **TOTAL** | **20** | | | |

response to the presence of potential competitors [8,54,55]. In our current study, the temporal sequence of detections of different individuals at the same locations reflected a system in which males monitor one another, and/or are attracted to the same shared resource (e.g., females, [53]). Overall, we infer that the trail system represents an important but limited spatial resource for males, not only used for traversing the forest, but specifically for monitoring competitors and finding receptive females.

The jaguar oestrus cycle is approximately 30–40 days [38], however postpartum lactational anestrus lasts 5–6 months [39]. During this period, it may be advantageous for males to kill unrelated cubs so as to return the female to cycling, as seen in other wild felids (e.g., lions, [28]; pumas, [19]; leopards, [27]). Instances of infanticide have been documented among jaguars, primarily in relatively open habitats where visibility is high and hiding cubs from infanticidal adults is difficult (e.g., Venezuela, [56]; Cerrado, [57]; Pantanal, [58]). Under these conditions, female jaguars display pseudo-oestrus and engage in non-conceptive matings, thereby confusing paternity and/or reinforcing pair-bonding as counterstrategy to infanticide [59,60]. In contrast, the dense forest of the Cockscomb Basin provides ideal cover for female jaguars with young to physically evade infanticidal males. This is corroborated by our previous work which has shown that females

appear only sporadically on the trail system frequented by males, and usually without dependents [34]. Previously, [35] suggested that jaguars in Cockscomb have a distinct mating season coinciding with the end of the wet season (November to January). This observation was based on one year of continuous camera trap data. However, in our current study, spanning 15 years, we documented close encounters between males and females year-round (Harmsen & Foster unpubl. data), suggesting that courtship and mating may occur at any time of the year, despite peaking towards the end of the wet season. Under such conditions, males cannot monopolise females, rather they must search for receptive females year-round [36]. If we assume stable female ranges; successful males will be those who become familiar with the ranges of multiple females, monitoring and moving between them as they come into oestrus. This hypothesis is supported by our findings that male activity centres shifted from one year to the next, and that males returned to females with whom they had associated in previous years.

Male activity centres formed temporary clusters with up to four males within 2 km of each other ('neighbours'). For most males, their activity centre and neighbours changed from one year to the next. We interpret this high level of flux and overlap as the response of multiple males to the appearance of receptive female(s) on the trail system: ranging widely when females are scarce on the trail system, and contracting when female presence increases [35]. Similarly, [20] found that among both pumas and snow leopards, males monitored several females but reduced their area of use to individual females when they were in oestrus.

Within the system of widely overlapping home ranges described here for Cockscomb, there is some evidence of dominance hierarchies governing space use in the area. This is illustrated by examples of 'take-overs' in which the abrupt disappearance of a male resulted in range shifts and increased use of the area by neighbouring males [31, 43, S2_File]. However, these observations remain consistent with the hypothesis that males monitor multiple females over a wide area and return to them when they come into oestrus. Compared to young males, prime males will be more experienced and familiar with the ranges of females, some of whom they may have previously encountered, mated or sired offspring. Marking by these prime males may deter weaker competitors. This is supported by our finding that old males encountered prime males less than expected. This could reflect the avoidance of strong, prime age competitors by the physically weaker and older age class. Notably, old males also had fewer close encounters than expected with females; potentially they avoid areas frequented by prime males and as a consequence have reduced access to females. Overall, the male reproductive strategy in our study areas appears to be 'relaxed' or non-territorial which switches to mate-guarding when females are encountered [20,30].

Male coalitions (of two jaguars) have been documented by [61] in the Pantanal and Llanos, associated with areas where female jaguars are highly concentrated. Under such conditions, males have been observed together: patrolling, marking, invading territories, and chasing off competitors, feeding on the same prey, and mating with the same female in the company of one another [61]. The locally high female densities at these sites are thought to be driven by the abundant prey that aggregate near waterbodies [61]. In contrast, in the dense forest of the Cockscomb Basin, we do not expect prey to aggregate, locally high densities of females, or the formation of male-male coalitions. Indeed, we found no direct evidence of coalitions in our study area, however, young adult male jaguars encountered one another more often than expected given their probability of detection along the trail system. This may reflect littermates associating with one another prior to dispersal, and is supported by our observations of two young male jaguars, identified as brothers from the same litter, travelling together (Harmsen & Wooldridge unpubl. Data, S2_File). We also documented more encounters than expected between young males and old males.

To our knowledge, this is the first long-term camera-trap study of jaguar socio-spatial dynamics to follow cohorts throughout their lives. As long as the study area is accessible, camera traps provide a cost-effective, non-invasive method for monitoring long-term space use of multiple individuals. We acknowledge that the camera grid only provides a crude representation of spatial range, limited by the extent of the camera array and density of camera stations; however, our metrics of overlap and crowdedness, and assessment of close encounters, allowed us to describe individual space use in relation to

conspecifics, and how this varied through time. The analytical methods are relatively simple to implement. We used close encounters as a proxy for interactions, which may represent direct meetings or visiting the same location within 24h, allowing us to assess the frequency with which different demographics of the sampled population met, avoided or investigated one another. Because of the dense closed habitat in which jaguars usually live, direct behavioural observations are generally difficult and rare. Even in the best-studied jaguar populations of the Pantanal, where direct observations are combined with GPS tracking data, data on dynamic interactions are limited to a handful of individuals (e.g., [13,62,63]). Our novel method opens the door for assessing space use and social systems from sparse camera trap datasets, which have historically been used for occupancy or abundance estimates only. We recognise the recent development of robust methods to assess spatio-temporal interactions between species using an occupancy framework within a single analysis (e.g., [64–66]), and the potential to apply it to studies of interactions between individuals. However, in a study of carnivore sympatry, Karanth et al. [65] noted that the method may be biased with low sample sizes. As such, it may be unsuitable for assessing interactions between individuals unless there are sufficient detections of every individual (across years).

Understanding limitations of a species' social plasticity, such as tolerance of crowding and overlap, will be informative for understanding carrying capacity when prey are not the limiting factor. With conservation management plans striving for ambitious goals such as doubling population numbers within existing or shrinking protected areas (e.g., tigers [67]), consideration should be given not only to whether there are sufficient prey and habitat, but also whether the species has the behavioural flexibility to live at these higher densities, wherein increased aggression could otherwise threaten recruitment. For example, in a highly competitive environment with limited refuges, such as open savannahs, increased density may cause territorial disputes and high turnover of males which can have disastrous consequences for cub survival (e.g., lions [68]). Low toleration for range overlap or crowding would put a lower limit on home range size, and thus density even in optimal prey conditions. Ultimately, density is a function of home range size, overlap, and crowdedness. Quantifying these parameters, as we have attempted here for overlap and crowdedness, will contribute to the better understanding of the limits to species population size.

## Supporting information

**S1 Fig. Annual activity centres for 11 male jaguars with ≥ 5 years of activity centres.**
(PDF)

**S2 Fig. Annual locations of activity centres for male jaguars with ≥ 5 detections per year.**
(PDF)

**S1 File. Supplementary methods.**
(PDF)

**S2 File. Supplementary Case Studies.**
(PDF)

## Acknowledgments

We thank the Belize Audubon Society for facilitating our research in the Cockscomb Basin Wildlife Sanctuary for almost two decades, and the Government of Belize's Forest Department for supporting our long-term work in Belize. For additional field assistance, we thank Miranda Davis, Said Gutierrez, Paul Higginbottom, Emiliano Pop, Emma Sanchez, Yahaira Urbina, Marvin Vasquez, Rebecca Wooldridge, Claudia Wultsch and Ferdie Yau. We thank Hugh Robinson for his constructive comments on the draft. This work is dedicated to memory of our good friend, Nicacio Coc, whose lifelong dedication to the Cockscomb Basin Wildlife Sanctuary will serve as an inspiration to the protected area managers of Belize for years to come.

## Author contributions

**Conceptualization:** Bart J. Harmsen, Rebecca J. Foster.

**Data curation:** Bart J. Harmsen.

**Formal analysis:** Bart J. Harmsen, Rebecca J. Foster.

**Investigation:** Bart J. Harmsen, Rebecca J. Foster.

**Methodology:** Bart J. Harmsen, Rebecca J. Foster.

**Project administration:** Bart J. Harmsen.

**Validation:** Rebecca J. Foster.

**Visualization:** Bart J. Harmsen, Rebecca J. Foster.

**Writing – original draft:** Bart J. Harmsen, Rebecca J. Foster.

**Writing – review & editing:** Bart J. Harmsen, Rebecca J. Foster.

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
