## [Decision Letter · Decision Letter 0]

6 Sep 2024

Dear Dr. Harmsen,

Thank you for submitting your manuscript to PLOS ONE. After careful consideration, we feel that it has merit but does not fully meet PLOS ONE’s publication criteria as it currently stands. Therefore, we invite you to submit a revised version of the manuscript that addresses the points raised during the review process.

We look forward to receiving your revised manuscript.

Kind regards,

Randeep Singh

Academic Editor

PLOS ONE

Journal requirements: 1. When submitting your revision, we need you to address these additional requirements. Please ensure that your manuscript meets PLOS ONE's style requirements, including those for file naming. The PLOS ONE style templates can be found at https://journals.plos.org/plosone/s/file?id=wjVg/PLOSOne_formatting_sample_main_body.pdf and https://journals.plos.org/plosone/s/file?id=ba62/PLOSOne_formatting_sample_title_authors_affiliations.pdf. 2. Thank you for stating the following in your Competing Interests section:  [No]. Please complete your Competing Interests on the online submission form to state any Competing Interests. If you have no competing interests, please state ""The authors have declared that no competing interests exist."", as detailed online in our guide for authors at http://journals.plos.org/plosone/s/submit-now  This information should be included in your cover letter; we will change the online submission form on your behalf. 3. Please provide a complete Data Availability Statement in the submission form, ensuring you include all necessary access information or a reason for why you are unable to make your data freely accessible. If your research concerns only data provided within your submission, please write "All data are in the manuscript and/or supporting information files" as your Data Availability Statement. 4. Please include captions for your Supporting Information files at the end of your manuscript, and update any in-text citations to match accordingly. Please see our Supporting Information guidelines for more information: http://journals.plos.org/plosone/s/supporting-information.

Reviewers' comments:

Reviewer's Responses to Questions

**Comments to the Author**

1. Is the manuscript technically sound, and do the data support the conclusions?

Reviewer #1: Yes

Reviewer #2: Yes

Reviewer #3: No

Reviewer #4: Yes

2. Has the statistical analysis been performed appropriately and rigorously?

Reviewer #1: No

Reviewer #2: Yes

Reviewer #3: No

Reviewer #4: Yes

3. Have the authors made all data underlying the findings in their manuscript fully available?

Reviewer #1: Yes

Reviewer #2: Yes

Reviewer #3: No

Reviewer #4: No

4. Is the manuscript presented in an intelligible fashion and written in standard English?

Reviewer #1: Yes

Reviewer #2: Yes

Reviewer #3: Yes

Reviewer #4: Yes

Reviewer #1: This study used a long-term camera trap dataset to assess socio-spatial dynamics of a jaguar population in Belize. This was a unique and effective way to use camera trap data to understand a population of an elusive species. I found the paper interesting to read, and it’s clear the authors put forth a great deal of work and effort into this study. The paper is light on statistical analyses, so it’s mostly descriptive, but that may be okay. I have the following detailed comments on the paper.

1. Abstract, line 19: Here you’re saying that males shared half of their range with another male. I see how you came up with half using your VI method in the SI, but using the distance between activity centers method, it seems like it’s more than half. See also my comment #24 below.

2. The first paragraph of the Introduction is long and could be broken up into two or more paragraphs. The first paragraph could be about large carnivores generally, and the second one about felids. The part beginning “In this study” (line 51) should be moved towards the end of the Introduction to the other paragraphs that discuss what you did in this study.

3. Lines 74-88: I don’t think you really need this paragraph, or at least, it could be condensed. This gives details about aggression during reproduction and infanticide of cubs, topics you did not focus on in your study. You also didn’t even find aggression in your study area, so this paragraph seems extraneous. Overall, the Introduction is long, and this is one place where you could cut some text.

4. Line 92: Did you mean to say “animals” or “individuals” here, instead of the first “leopards”?

5. Line 144: I’m not sure if you should include #5 here in this list since it was only discussed in the Supplementary Information and not in the main text.

6. Line 180: Did the study begin in 2002 or 2003? Here you say 2002, but in several other places throughout the text, you say 2003. These just need to be consistent.

7. Line 193: How often did you check the cameras (i.e., change batteries, switch out film or SD cards, etc.)?

8. Line 258: I’m confused why this is a paired Wilcoxon test. If the sampling unit is the sequence (either same or diff), and you would not expect to have the same number of same sequences and diff sequences, then what exactly is being paired?

9. Lines 268-273: I think you have these backwards. A left skew is when the tail of the distribution is longer on the left side, so the bulk of the observations are large, and a right skew is when the tail is longer on the right side, so the bulk of the observations are small. In your case, you found the most observations in the shorter-interval bins (Figure 2), so this would be a right-skewed distribution.

10. Also, is there any way to test “evidence for attraction” that is more quantitative? This method seems to rely solely upon visual inspection of a plot, which is not statistically rigorous.

11. Lines 286-287: How did you determine the age of each jaguar from camera images?

12. Line 290: I’m not following how you calculated expected detection probability of the dyad. If the dyad was two different age classes (e.g., young-old), you would have two different probabilities of detection, one for young and one for old, so did you multiply them to get the expected detection probability of the dyad?

13. Line 307: This would be right-skewed. Also, same comment as before – visual inspection of a plot is not very quantitative.

14. Line 319: Are you assuming that every male is less than 2 years old when first detected? What if it’s an older male that simply evaded the cameras in previous years? Also, is there any immigration into your study area? I think you need to better explain these age determinations.

15. Line 326: Please name the correlation test you did. Your paper is light on statistical analysis, so when you did perform a test, you should name it and the software used (I’m assuming it’s R).

16. Line 332: For the number of close encounters with males, especially within the 20-day period, did you still exclude encounters with the same male in the same week (as described on lines 315-316)? You don’t use the word “independent” in this paragraph, so it’s not clear. If you did exclude them, that would limit the number of encounters with that male to 3 in the 20-day period. If you counted all of them for this analysis, please clarify that.

17. Line 351: I’m not sure you can say this without statistically testing for a shift over time.

18. Line 379: I thought you only used males with at least 5 years of data for this analysis (from line 238), but here you say 4 years.

19. Lines 383-384: This is confusing. You just said above that 13 males maintained the same neighbors for 2 years, now you’re saying it’s 4 individuals. What are you saying differently in this sentence? I also don’t know what you mean by “all were in different years”.

20. Lines 392-393: What do “3d Q = 10” and “3d Q = 6” mean?

21. Table 1: The caption should be changed to “Frequency of observed and expected encounters between male and female jaguars, according to male age class (prime, 3-7y; old, 8+y)…”. You shouldn’t have “close” encounters and the definition of it in this first sentence because panel a is “close” and panel b is “true”.

22. Line 466: This says “over multiple years”, but in the Methods (lines 328-329), you say that you recorded the actual number of years over which males repeatedly encountered the same female. Can you provide the actual number of years?

23. Line 467: Do you mean maximum annual “distance moved”?

24. Line 499: Not sure where you’re getting “half” of their range. You reported that male activity centers were, on average, within 2 km of other males, so wouldn’t this be more than half?

25. Lines 511-522: As in the Introduction, I don’t think this discussion of infanticide really adds anything to your paper.

26. Line 532: If female ranges are stable, and males associate with the same females each year, then why would this explain male activity centers shifting from year to year?

27. Line 534: Four other males? On line 373, you say the maximum number of neighbors was 3.

28. Line 570: Any thoughts on why young and old males might have more encounters than expected?

29. Figure 1: It’s difficult to see the difference between the access road and the trail. Could you use a different type of line for the access road, such as dashed? Also, are there 20 gray circles? I’m only counting 19.

30. Figure 2: For this figure and all others that show boxplots, please explain each part of the box, including the whiskers and outlier points.

31. Figure 5: The caption for this figure and the text in the Results both say that n = 237 close encounters, but the figure’s x-axis says that n = 474. The numbers in each bin seem to add to 474, so is this figure incorrect?

Supplementary Methods

32. Line 6: Why only 7 detections? Kernel home ranges should usually have more detections than that.

33. Lines 19-20: Unclear sentence.

34. S1, Associations between conspecifics: In Case 1, you’re referencing Figure S3; I don’t see that figure. In Case 3, you say that the activity center in 2012 for M02-5 is not shown in Figure S1, but there is one shown in that year.

35. Figure S2: Looks like the legend is not on 2017’s figure, but is included in multiple other years.

Reviewer #2: PONE-D-24-29955

Thank you for the opportunity to review this manuscript. The following comments are intended as a constructive criticism to improve the quality and scientific value of their contribution to the scientific literature. The manuscript is a long-term camera trap monitoring of 15 years with a robust dataset in a protected area in Belize (Central America) and it assessed the socio-spatial interactions and dynamics such as territoriality and range shifts in a high-density jaguar (Panthera onca) population. Authors recorded 136 individuals during the study period and focused the analysis on overlapping and crowdedness among males (~4,000 independent records), as they considered the detections of females (~700 independent records) insufficient to analyze some interactions and ranges throughout the years but included this data to assess the close encounters (sequential records of different individuals at the same location within 24 hours), adding the age of males as a variable. The authors acknowledge the weaknesses of the manuscript, such as potential biases caused by installing cameras only on trails (biasing the records towards males) and that the home ranges of some individual jaguars extend outside the camera grid (~100 km2). The tests were simple and replicable for other study areas. Although the manuscript presents some weaknesses and potential biases due to the much more elusive behavior and unpredictability of females’ movements, which decrease the detection probability in camera traps, the authors provided a non-invasive and cost-effective method for monitoring long-term space use of cryptic species. In general, the manuscript is very well written and with adequate literature, and I do recommend it for publication in PLoS One. However, I have minor comments that should be addressed before accepting it.

I offer comments on each section below:

Abstract:

Line 17: change “8 km” for “7.7 km”.

Introduction:

Lines 61-63: Although the number of mating events is important, it is not sufficient to state that it represents the reproductive success of males, as only a few mating events result in fertilization in this species. Please, rephrase this sentence.

Lines 63-67: Authors can include a short sentence relating these differences between males and females’ movements with the male-biased dispersion found for this species.

Lines 114-115: change “Ims (1987) for “(36)”.

Line 144: change “take-overs” for “takeovers”.

Methods:

Line 170: although the density estimates are provided in the cited literature, I suggest mentioning this information in the text to make it easier to read.

Line 171: Add “.” at the end of the sentence.

Line 174: during the entire manuscript the study period was between 2002 to 2017. I think here it was a typo. If yes, change “2003” for “2002”.

Line 180: is this camera trap grid size enough to encompass home range of males? Are there any estimates of home range sizes for males and females for this population? Please provide this information.

Line 190: the 3-minute delay between exposures is understandable due to the equipment limitations at that time (2003-2008), but it is important to mention in the results section that this delay may have underestimated sequential records of different individuals at the same camera location.

Lines 215-216: please provide a reference to support this or clarify why you chose 7 detections per year as your baseline.

Line 233: why did you choose ≤2 km as your baseline? Please explain.

Line 238: why did you choose ≤2 km as your baseline? Please explain.

Line 252: indifference or tolerance?

Line 258: replace “r” for “R”

Line 267: why did you choose 15 ‘diff’ sequences as your baseline? Please explain.

Lines 286-287: even mentioning the criteria used to estimate the ages in the cited study (Harmsen et al. 2017), it is

not clear how the authors estimate the ages of adult males that appeared in the population over the years, specially in the limits of age classes. Did you consider characteristics such as body size and body conditions?

Line 301: I did not find this information (5.5 km of home range radius for males) in the cited work. Please check if this is information is accurate.

Lines 310-311: although the detections of females (700 independent records) were considerably lower compared to males (4,000 independent records), have you tried using the same tests as you used for males? I am curious about this possibility.

Lines 314-315: why did you choose ≥ 1 week as your baseline? Is it related to the average mating period in the study area? Please explain.

Lines 332-333: the original study that mention this information is “Almeida, A. E. 1976. Jaguar hunting in Mato Grosso. England, Stanwill Press”. Please, add it to the reference list. I also suggest checking more recent studies to be sure if they support such a long mating period for jaguars.

Results

Line 366: add “.” at the end of the sentence.

Line 371: please provide the range.

Line 399: please replace “16-year study period” for “15-year study period” as mentioned in other sections of the

manuscript.

Line 467-470: there is only one sentence in this paragraph. I suggest putting this paragraph together with the previous one.

Line 529: cite a reference to support the stability of female ranges

Figures:

Figure 01: I see no difference between the colors of the trail system and the roads. Perhaps use dotted lines for trails to better differentiate them.

Reviewer #3: This paper aims to describe the socio-spatial dynamics of jaguars in a forested landscape of Belize. The study suffers from a small sample size of camera-trap locations, while the additional cameras are not well documented. More information is needed on these additional cameras in order to assess the validity of results. Some of the stats need checking. I do not believe the data can be used to assess territoriality- behavioral observations or locations from radiocollars are necessary. The Discussion makes some sweeping statements that are unwarranted.

Specific Comments:

L. 25 “Notably, old males also had fewer close encounters than expected with females”- this is not shown by the data.

L. 188-195 “From 2003 to 2008, we used traditional film camera traps (CamTrakker, Cuddeback and DeerCam) with an enforced 3-minute delay between exposures to prevent excessive photos of herding species such as peccaries (Tayassu sp.). From 2011 onwards, we used digital camera traps (Pantheracam; Panthera corp, New York, NY) with a minimum delay of 8 seconds between successive photo triggers. Every photograph was stamped with the time and date. We identified individual jaguars based on their unique spot patterns, and assigned sex based on the presence or absence of testicles, following (46).”- The wording here is exactly the same as in a previous publication (Harmsen et al 2017). Is this self-plagiarism? I suggest the wording be altered.

L. 194 “We identified individual jaguars based on their unique spot patterns”- It would be useful to know what proportion of jaguar detections were identified to an individual and what proportion were unidentified.

L. 280-282 “If males proactively defend exclusive territories then the probability of close

encounters between prime males would be higher than the random expectation (attraction at sites of territorial disputes)”- I’m not sure this is true in the present field situation. One would expect that encounters would be high only at the edge of territories but low in the center of defended territories. It is unclear to me how one determines the edge of a territory using only 20 camera-trap locations. Also, from Figure S2, it looks like each male was only picked up at 5 to 6 camera traps if their range was about 10 km across. Harmsen et al 2020 concludes that a higher density of camera-trap locations is necessary for reliable density estimates due to variation in behavior. L. 187-188 does state “with additional data from small-scale camera- and video-trap surveys deployed inside and outside of the sanctuary, resulting in a database of 210 monitored locations”- but the range of dates when these additional camera traps were used is not specified. This is necessary to assess the robustness of the data.

L. 287, L. 320-321 Age was assessed according to years of detection and thus could easily have been underestimated in the early years (2002, 2003 and again in 2011, 2012). The previous article (“34”- Harmsen et al 2017) states “For each year, we calculated the minimum age of the individuals detected, using their year of first and last detection. We assumed that adults were at least 2 years old on first solo detection”. This means that animals assessed to be old were definitely old, but those assessed as young in the early years could have been prime-age or old. Also, prime-aged animals could have been old. This bias needs to be pointed out, with the assumption that animals were not alive before (or after) they were sighted on the study area.

L. 326, 327, 331, 332 Use square brackets for the references, not here, where they should be parentheses.

L. 18, 356, 357, 417, 420, 443, 465, 470 “N”- be consistent in use of “n”.

L. 359-361 “Maximum moves in activity centre occurred at different periods during tenure, with two individuals moving maximally at the start of their tenure, two at the end, and the remainder during the middle (Supplementary Fig 1).”- The greatest moves of >12 km all encompass moves between years when data were missing (2008-2011, 2007-2012) and so were likely not large annual moves. Moves with missing years should be excluded from analyses. Most males seemed not to move much from one year to the next, with the greatest move of about 6 or 7 km.

L. 358, 359 “with nine individuals shifting their activity centre by more than 5 km from one year to the next.”- is this correct, excluding the missing years?

L. 382, 383 “illustrating a high level of flux in range use and neighbours”- but were the neighboring males alive the following year or is this flux more to do with mortality? Clarify at L. 238 whether this is for pairs of males that were both alive each year.

L. 392, 393 “(3d Q = 10)” and “(3d Q = 6)”- Unclear meaning for Q. Also, is the test statistic for Wilcoxon’s signed rank test W, not V?

L. 412, 414, 416, 418 “proportion of … encounters”- strictly speaking, it is the “number of … encounters” that you test with a Chi-squared test.

Figs 3 and 4- How did you arrive at the level of significance of p<0.001? This seems very low for Fig 4 where the sample size is small (n = 32). I believe the result of a Chi-squared test on these data would have p-value of about 0.16. In fact, the Chi-squared test would be invalid because more than 20% of the cells have a value less than 5. A Fisher’s Exact test would be p = 0.15. This means the number of true encounters for all age-class combinations was as expected. As for Fig 3, Chi-squared would be about 16.5 with p = 0.0056. Please check your calculations. In addition, the Chi-squared test with 5 df only gives a p-value overall, not for specific age-class combinations in both cases.

L. 454, 455. The stats seem incorrect here. Chi-squared on the close encounters = 2.99, p = 0.08 and for the true encounters, Fisher Exact test, p = 0.09. The encounter frequency is not different according to age of the male.

L. 485 “rich data”- I would not say that the data are rich, but sparse. I realize that jaguars are an elusive species, but rich data require large data sets (many camera stations). As you state on L. 104, 105 “camera trapping will only provide a crude representation of spatial range”.

L. 531, 532 “This hypothesis is supported by our findings that male activity centres shifted from one year to the next”- if males are familiar with resident females, then male ranges should not shift, but be large enough to encompass several female ranges.

L. 532, 533 “males returned to females with whom they had associated in previous years.”- do your data actually show that males returned to specific females? The Results do not clearly show this, only that “five males 46% had repeated encounters with the same female(s) over multiple years” L. 466.

L. 534 “Male activity centres formed temporary clusters”- I wonder if the clustering of male activity centers is not simply the result of few camera stations that were set out linearly, so the centroids must be near each other. If the camera stations were more evenly spread over the landscape, then the centroids would also be more spread out.

L. 549 “off-spring”- has no hyphen.

L. 553 “Notably, old males also had fewer close encounters than expected with females”- This was not the case. Old and prime males encountered females as expected.

L. 572 “to follow entire cohorts throughout their lives”- if this sweeping statement is true, then you need to show that 1) the camera traps did not miss any individuals, i.e., all were identified and 2) that individuals were followed from maturity to death. How long do males usually live? L. 320 has a mean of 3.1 annual activity centers per individual (range 1 to 11). Do males only live to 4 years on average (followed at age 2, 3, 4)? I would guess it would be more like 12 years. This means that you would have been able to follow only a couple of complete cohorts. And then there was a 3-year gap in the study from 2008-2011.

L. 599, 600 “Ultimately, density is a function of home range size, overlap, and crowdedness. Quantifying these parameters, as we have attempted here”- I do not think your data are appropriate for quantifiying home range size or overlap.

Fig S2 “(for legend see year 2017)”- what does this mean? There is no legend for 2017.

In several places, references in the text do not follow the citation-sequence format, e.g. L. 114-115.

I can not see the Excel file of raw data.

Reviewer #4: This article addresses the social and spatial dynamics of a jaguar population in a protected area in Belize, Central America. This is a long-term study (15 years long) with consistent camera trap data. The main findings are that males were not territorial and their range changed constantly over the study period; males shared their range with other males; and old males encountered young males and females less than expected, indicating the old jaguars avoid competing with stronger young jaguars. This is a well-written manuscript with consistent data and interesting results. A constraint in the methods is the use of sequences of records of the same and different jaguars as a proxy for interactions. Camera traps were also clumped on trails and the authors may consider that the overlapping of jaguar movement may be overestimated. The authors may explicitly present the reasoning or evidence to give support for the assumptions about choices like: selection of individuals recorded 15 times, taken as independent records those with time intervals higher than one week; fifteen records of different jaguars over time. Inferences about jaguar behavior depends on consistent methodological approach and so you must provide a consistent reasoning or evidences to support definitions. In the discussion, the authors also may rethink about the determinants of population size. Behavior is very important, but prey abundance may be a higher constraint in the case of jaguars as found in the literature.

Lines

Abstract

18 - Correct the +/- expression to an adequate format

19-20 - The phrase is a little hard to follow. Please reword.

23 - Must be >= 8 y.

Introduction

51-57 This allusion to your study seems to be adequate in the end of the Introduction.

77-78 This sentence is like a list. Please, reword.

89-90 You will conclude something using these data: Consider “long-term data can be used to address” or something similar

95 Is there few or no studies?

112 - 119 This part of the paragraph is disconnected from the above initial sentences because subtly you started about theory, but before you started describing your study site. Try to start with theory in the beginning of the paragraph.

141-142 Is the degree of overlap an indicator (proxy) for evidence of avoidance, attraction and tolerance? If yes, I suppose the goal of estimating overlap is to infer about avoidance, attraction and tolerance. Thus you have only one objective, which is repeated with the indicator variable (overlap).

156-157 You better use this phrase in the beginning of discussion.

Methods

Did you tested spatial autocorrelation? The camera-trap stations look clumped in the east part.

206-210 Don't you have some information about jaguar movement outside trails. As cameras are limited to trails, old males and females may change their patterns of movement, avoiding trails, which would be an explanation to the low encounter rates.

240-245 Variability in home range can also be interpreted as an indication of competition, so that a jaguar may reduce its range for defense, or increase its home range to gain advantage over an adjacent jaguar’s territory.

250-252 This phrase is truncated. Please reword.

258 Use capital letter for 'R' Software.

258-262 What if only one jaguar is detected by the camera station? Did you label it as avoidance? Is there such a situation? If there is no other jaguar around, a long sequence of the same jaguar, would not mean that it is avoidance, nor any interaction within active centers?

267 Why fifteen?

264-273 Is there any evidence to support your reasoning? Could not a jaguar be in a rush to avoid others and cross in front of a camera trap in shorter time periods than expected by chance?

278 Why 24 hours?

282-284 You assume that old and young do not compete with prime. However, categories of old and young may not reflect the vigor of the animal. A 7-year-old animal can be weaker than and 8-year-old animal, or a 3-year-old animal may have been in a bad condition than a 2-year-old.

315 Why one week?

Results

340 and 348 Standardize the format to report standard deviation (SD= or mean ± SD)

348 Please, standardize the format of SD.

356-366 No need to repeat N value as it was stated in the beginning of the sentence.

379 - 380 I suppose this male has appeared repeatedly at the camera records and has been classified as having avoidance. However, doesn’t it mean it has none interaction?

391-393 Please, try to express statistics in a consistent format through the manuscript.

422-430 Figs 3 and 4 can be put together in the same figure.

438 - 440 Isn't this discussion?

454-455 What was the probability? Show the exact value with as many decimals as shown above.

466 Correct the insertion of percentage. It may go between parenthesis.

479 Is the SD of zero correct?

Discussion

496-497 Is there any evidence of high prey abundance in this forest? Rainforests on well-drained soils of in Central Amazon have high richness of prey but abundance is low, whereas the Pantanal has a lower richness, but higher abundance.

495-510 How can you control for the effect of the trail system on the patterns you found? High overlapping may be an artifact of exhaustive sample on trails. As trails are treated as resource the overlap may be overestimated.

519-520 Is there any evidence of a relationship between vegetation density and infanticide in forest ecosystems? Or infanticide is just hard to be recorded in closed forest?

578-581 It would be clarifying if you provide a reasoning or evidence supportive to the idea that interactions within 24h are interactions.

594-597 Prey resources may be a significant constraint on population density at different habitats. For example: in a fragmented landscape in Mexico, Pantanal ranch and protected areas and in a small sea island in the Amazonian coast:

Luja VH, Guzmán-Báez DJ, Nájera O, Vega-Frutis R. Jaguars in the matrix: population, prey abundance and land-cover change in a fragmented landscape in western Mexico. Oryx. 2022;56(4):546-554. doi:10.1017/S0030605321001617

Devlin, Allison L., Jacqueline L. Frair, Peter G. Crawshaw Jr, Luke T. B. Hunter, Fernando R. Tortato, Rafael Hoogesteijn, Nathaniel Robinson, Hugh S. Robinson, Howard B. Quigley 2023 Drivers of large carnivore density in non-hunted, multi-use landscapes. Conservation Science in Practice https://doi.org/10.1111/csp2.12745

Duarte, H.O.B., Carvalho, W.D., de Toledo, J.J. et al. Big cats like water: occupancy patterns of jaguar in a unique and insular Brazilian Amazon ecosystem. Mamm Res 68, 263–271 (2023). https://doi.org/10.1007/s13364-023-00681-7

**Do you want your identity to be public for this peer review?** For information about this choice, including consent withdrawal, please see our Privacy Policy

Reviewer #1: No

Reviewer #2: No

Reviewer #3: No

Reviewer #4: No

---

## [Author Response · Author response to Decision Letter 1]

12 Mar 2025

Below you can find our rebuttal to the very helpful comments and questions of the 4 reviewers. Many thanks for your careful reviews, really appreciate. We think we provided adequate answers and made the necessary changes. Many thanks for considering our highly improved manuscript.

In the replies we have provided the numbered lines of where you can find the changes following the new manuscript without the track changes called BHarmsen_Manuscript. Our replies start with an Asterix

Reviewer #1: This study used a long-term camera trap dataset to assess socio-spatial dynamics of a jaguar population in Belize. This was a unique and effective way to use camera trap data to understand a population of an elusive species. I found the paper interesting to read, and it’s clear the authors put forth a great deal of work and effort into this study. The paper is light on statistical analyses, so it’s mostly descriptive, but that may be okay. I have the following detailed comments on the paper.

Many thanks for the encouraging statement

1. Abstract, line 19: Here you’re saying that males shared half of their range with another male. I see how you came up with half using your VI method in the SI, but using the distance between activity centers method, it seems like it’s more than half. See also my comment #24 below.

*This is a good point, we added that the source of half their range came from the VI method, while keeping the following distance statement regarding activity centres. Although we agree that the distance analyses might indicate higher overlap, we prefer to keep this to the distance unit to avoid metric confusion.

It now reads:

Male jaguars overlapped extensively with one another: per year, males shared, on average, half of their detected range with at least one other male (Volume of Intersect, kernel overlap); and their activity centres lay within 2 km of at least one other male. Lines 18-21:

2. The first paragraph of the Introduction is long and could be broken up into two or more paragraphs. The first paragraph could be about large carnivores generally, and the second one about felids. The part beginning “In this study” (line 51) should be moved towards the end of the Introduction to the other paragraphs that discuss what you did in this study.

*We have shortened the first paragraph now reading:

Conservation concern for populations of large carnivores has been widely acknowledged (1). Social factors play an important role in regulating the population dynamics of carnivores (e.g. 2), therefore, an understanding of carnivore social organisation can inform their management and conservation (e.g. 3). In this study, we demonstrate the utility of camera trap technology for studying the long term space use patterns and dynamic interactions of cryptic carnivores. Lines 36-40.

We equally made an additional paragraph cut between Lines 50 and 51

3. Lines 74-88: I don’t think you really need this paragraph, or at least, it could be condensed. This gives details about aggression during reproduction and infanticide of cubs, topics you did not focus on in your study. You also didn’t even find aggression in your study area, so this paragraph seems extraneous. Overall, the Introduction is long, and this is one place where you could cut some text.

*We consider it relevant for explaining why females avoid males; and why some males may associate with one another- topics that we refer to in Discussion

4. Line 92: Did you mean to say “animals” or “individuals” here, instead of the first “leopards”?

*We have changed “leopards” to “individuals” Line 93

5. Line 144: I’m not sure if you should include #5 here in this list since it was only discussed in the Supplementary Information and not in the main text.

*We have removed #5, Line 149

6. Line 180: Did the study begin in 2002 or 2003? Here you say 2002, but in several other places throughout the text, you say 2003. These just need to be consistent.

*We have corrected 2003 to 2002, Line: 180, 197

7. Line 193: How often did you check the cameras (i.e., change batteries, switch out film or SD cards, etc.)?

*We have added the following:

We checked and changed cameras, batteries, film and SD cards as required every 2-4 weeks depending on the location and the camera model. Lines: 205-206

8. Line 258: I’m confused why this is a paired Wilcoxon test. If the sampling unit is the sequence (either same or diff), and you would not expect to have the same number of same sequences and diff sequences, then what exactly is being paired?

*Because the number of jaguars detected varied between camera stations, we used ‘station’ as the paired unit of comparison. To clarify, changed the text accordingly:

We compared the number of ‘same’ sequences versus ‘diff’ sequences, per camera location, using the paired Wilcoxon test in R. Lines: 279-280.

9. Lines 268-273: I think you have these backwards. A left skew is when the tail of the distribution is longer on the left side, so the bulk of the observations are large, and a right skew is when the tail is longer on the right side, so the bulk of the observations are small. In your case, you found the most observations in the shorter-interval bins (Figure 2), so this would be a right-skewed distribution.

*Many thanks for this indication, we have changed accordingly in the text in the methods and results. Line 290-291 and Line 441

10. Also, is there any way to test “evidence for attraction” that is more quantitative? This method seems to rely solely upon visual inspection of a plot, which is not statistically rigorous.

*We included the result of the Shapiro-Wilk test for normality, indicating a significant deviation from a normal distribution. Line 300, Lines 483-484.

11. Lines 286-287: How did you determine the age of each jaguar from camera images?

*As we have long-term records, we determined age based on the length of time since the date of first detection.

Age was calculated as number of years since first detection + 2, assuming that individuals were at least 2 years old when first detected as an adult. We recognise the bias of underestimating age if certain individuals were first detected when they were older than 2 years. However, our long-term data show that first detection is mostly at a young age, supported evidence of young immigrants from neighbouring study areas establishing ranges at a young age in our study area (BHarmsen Unpubl. data). We always validated age class of first detection based on photographs, subjectively assigning age class based on head width, muscularity and testicular size Lines 308-315.

12. Line 290: I’m not following how you calculated expected detection probability of the dyad. If the dyad was two different age classes (e.g., young-old), you would have two different probabilities of detection, one for young and one for old, so did you multiply them to get the expected detection probability of the dyad?

*Yes, we multiplied the probability of detection of each age class of the dyad in order to calculate the probability of detection of an encounter between the two age classes

13. Line 307: This would be right-skewed. Also, same comment as before – visual inspection of a plot is not very quantitative.

*Similar to previous we changed to right skewed, as per suggestion and performed a Shapiro-Wilk test to assess deviation from normal distribution. Lines 342-343, Lines 483-484

14. Line 319: Are you assuming that every male is less than 2 years old when first detected? What if it’s an older male that simply evaded the cameras in previous years? Also, is there any immigration into your study area? I think you need to better explain these age determinations.

*See comment 11 -we explained how and why we assigned the age class. Now reading:

Age was calculated as number of years since first detection + 2, assuming that individuals were at least 2 years old when first detected as an adult. We recognise the bias of underestimating age if certain individuals were first detected when they were older than 2 years. However, our long-term data show that first detection is mostly at a young age, supported evidence of young immigrants from neighbouring study areas establishing ranges at a young age in our study area (BHarmsen Unpubl. data). We always validated age class of first detection based on photographs, subjectively assigning age class based on head width, muscularity and testicular size. Lines 308-315

15. Line 326: Please name the correlation test you did. Your paper is light on statistical analysis, so when you did perform a test, you should name it and the software used (I’m assuming it’s R).

*We indicated we used Pearson correlations, by adding:

….the maximum distance a male moved between years across its lifetime, using Pearson correlation (48). Line 364

16. Line 332: For the number of close encounters with males, especially within the 20-day period, did you still exclude encounters with the same male in the same week (as described on lines 315-316)? You don’t use the word “independent” in this paragraph, so it’s not clear. If you did exclude them, that would limit the number of encounters with that male to 3 in the 20-day period. If you counted all of them for this analysis, please clarify that.

*We added the word ‘independent’ to indicate we would not double count male encounters within the 20-day period. Line 367

17. Line 351: I’m not sure you can say this without statistically testing for a shift over time.

*There are movements, as we detected shifts in activity centres >0. We changed the word ‘suggesting’ and replaced it with ‘meaning’ to make it a statement that there is considerable movement on average. Line 391

18. Line 379: I thought you only used males with at least 5 years of data for this analysis (from line 238), but here you say 4 years.

*Many thanks, we have corrected this to 5, Line 424

19. Lines 383-384: This is confusing. You just said above that 13 males maintained the same neighbors for 2 years, now you’re saying it’s 4 individuals. What are you saying differently in this sentence? I also don’t know what you mean by “all were in different years”.

*We clarified this in the following manner:

Dyads in which males were neighbours (activity centres ≤ 2 km apart) had a low level of permanency through time, never exceeding 2 years and none of these two years concerned two consecutive years in a row (always staggered with other neighbour dyads in between). For the 22 males for which we could assess activity centres across multiple years (≥ 5 years, maximum 11 years), 59% (13/22) had the same neighbour for 2 years, 36% (8/22) had different neighbours each year, and one never had a neighbour; illustrating a high level of flux in range use and neighbours. Lines 421-427

20. Lines 392-393: What do “3d Q = 10” and “3d Q = 6” mean?

*It means the 3rd quarter of the ranking, as a standard reported output of the Wilcox test in r.

21. Table 1: The caption should be changed to “Frequency of observed and expected encounters between male and female jaguars, according to male age class (prime, 3-7y; old, 8+y)…”. You shouldn’t have “close” encounters and the definition of it in this first sentence because panel a is “close” and panel b is “true”.

*We have corrected the caption as requested, Lines 501-504

22. Line 466: This says “over multiple years”, but in the Methods (lines 328-329), you say that you recorded the actual number of years over which males repeatedly encountered the same female. Can you provide the actual number of years?

*We have now provided the actual numbers, the sentence now reads:

Of these, five males (46%) had repeated encounters with the same female individual(s), frequently across multiple years (Mean = 1.6 years, SD = 0.8, range 1-3). Lines 509-511

23. Line 467: Do you mean maximum annual “distance moved”?

*We changed to ‘maximum annual distance moved’, Line 512

24. Line 499: Not sure where you’re getting “half” of their range. You reported that male activity centers were, on average, within 2 km of other males, so wouldn’t this be more than half?

*We changed this to:

Indeed, males overlapped extensively in space, sharing at least half of their detected annual range with at least one other male. Lines 546-547

25. Lines 511-522: As in the Introduction, I don’t think this discussion of infanticide really adds anything to your paper.

*After careful consideration we still think it is needed to explain why females avoid males and males must run all over the place looking for them. We agree that this is the section that has the least direct connection to our data, but we prefer to keep it.

26. Line 532: If female ranges are stable, and males associate with the same females each year, then why would this explain male activity centers shifting from year to year?

*They associate with the same female but they do not associate with the same female every year. As females become unavailable for mating for the 2 years when they raise young, they look for other females and shift their range accordingly. They check the same females but shift activity focussing on different receptive females, which are available in different parts of the area they cover at different times.

27. Line 534: Four other males? On line 373, you say the maximum number of neighbors was 3.

*One individual with 3 neighbours makes 4 individuals. We removed the word other to make this clearer. Line 584

28. Line 570: Any thoughts on why young and old males might have more encounters than expected?

*We supposed that these dyads could represent father-son dyads, however in the absence of strong evidence we decided that this was too speculative to bring into the Discussion.

29. Figure 1: It’s difficult to see the difference between the access road and the trail. Could you use a different type of line for the access road, such as dashed? Also, are there 20 gray circles? I’m only counting 19.

*We have changed the grey scale and made the line thicker to distinguish the access road from the trail system and park boundary

There are 20 camera locations, two are however relatively close together

30. Figure 2: For this figure and all others that show boxplots, please explain each part of the box, including the whiskers and outlier points.

*We have now included the explanation in the legend of the various parts of the boxplot, now reading:

Fig 2 Proportion of consecutive detections of individual X with any other individual within 1, 2, 3 … 30(+) days of each other; mean (horizontal bars) shown for 37 individuals, the means (black line in the grey box) of all bins sum to 1, the grey box represents 50% of all the data on either side of the mean with the whisker lines extending outside of the grey box, showing the maximum and minimum range, excluding outliers 1.5* interquartile distance from the median. Outlier points outside of this range are indicated as points above and below the whisker lines. Lines 449-455

31. Figure 5: The caption for this figure and the text in the Results both say that n = 237 close encounters, but the figure’s x-axis says that n = 474. The numbers in each bin seem to add to 474, so is this figure incorrect?

*Many thanks for noticing. This analysis ran with interactions doubled (237 ×2 = 474) with all interactions included per individual, making that dyads were doubled. However, this has no impact on the outcome. We reran the analysis with single dyads to assure the sample size of dyads was reduced to the original number of 237.

With the rearrangement of Figures, per other comments, Figure 5 is now Figure 4. Caption for this figure now reads:

Fig 4 Frequency of close encounters between 37 male jaguars according to the distance between their activity centres, n = 237 close encounters. Lines 486-487

Supplementary Methods

32. Line 6: Why only 7 detections? Kernel home ranges should usually have more detections than that.

*It is the minimum number of points to allow the VI kernel calculation; We added the mean number of locations and detections, with SD used for kernel calculation, indicating most kernels are based on much larger number of detections.

For each jaguar with at least 7 detec

---

## [Decision Letter · Decision Letter 1]

27 May 2025

Dear Dr. Harmsen,

Thank you for submitting your manuscript to PLOS ONE. After careful consideration, we feel that it has merit but does not fully meet PLOS ONE’s publication criteria as it currently stands. Therefore, we invite you to submit a revised version of the manuscript that addresses the points raised during the review process.

We look forward to receiving your revised manuscript.

Kind regards,

Randeep Singh

Academic Editor

PLOS ONE

Journal Requirements:

Additional Editor Comments:

<!--StartFragmentI appreciate your effort and recommend to carry revisions to the manuscript.<!--EndFragment

Reviewers' comments:

Reviewer's Responses to Questions

**Comments to the Author**

Reviewer #5: (No Response)

2. Is the manuscript technically sound, and do the data support the conclusions?

Reviewer #5: Yes

3. Has the statistical analysis been performed appropriately and rigorously?

Reviewer #5: Yes

4. Have the authors made all data underlying the findings in their manuscript fully available?

Reviewer #5: Yes

5. Is the manuscript presented in an intelligible fashion and written in standard English?

Reviewer #5: Yes

Reviewer #5: The manuscript “Long-term spatial dynamics of jaguars in a high-density population” brings us an excellent analysis of the socio-spatial dynamics of a jaguar population in Belize. I consider the manuscript to be very well written and to have a good statistical approach to the objectives described. After reading the article and the comments that had already been made by three other reviewers in rounds prior to my evaluation, my only suggestions would be these:

1) reduce the introduction by 10-20% as it is too long

2) In this paper, the authors could have considered doing spatiotemporal analysis, not just separating spatial and temporal analysis. Please see, for example, the papers of Karanth et al. (2017), Pereira et al. (2024) and Duarte et al. (2025) to get an idea of a possible spatiotemporal approach that considers both the sites of the cameras and the time at which the photo or video was taken, which would give us an idea of whether there is a temporal exclusion between males and females at each of the sampled site. I understand that this would not be the objective of the present study, and the authors can consider this in a future analysis. However, perhaps the authors could mention this in the discussion, considering it as a future approach that could shed more light on socio-spatial dynamics issues.

References

Duarte HO, Rosalino LM, Toledo JJ, Hilário RR, Carvalho WD. 2024. Spatiotemporal interactions between jaguars (Panthera onca) and their potential prey in Amazonian islands. Ecological Research 40(2):217-227. https://doi.org/10.1111/1440-1703.12522

Karanth KU, Srivathsa A, Vasudev D, Puri M, Parameshwaran R, Kumar NS. 2017. Spatio-temporal interactions facilitate large carnivore sympatry across a resource gradient. Proceedings of the Royal Society B: Biological Sciences 284(1848):20161860. https://doi.org/10.1098/rspb.2016.1860

Pereira R, Matias G, Santos-Reis M, Rosalino LM. 2024. Influence of habitat edges on spatial and spatio-temporal occurrence patterns of mesocarnivores in landscapes dominated by Eucalyptus plantations. Forest Ecology and Management, 572, 122257. https://doi.org/10.1016/j.foreco.2024.122257

**Do you want your identity to be public for this peer review?** For information about this choice, including consent withdrawal, please see our Privacy Policy

Reviewer #5: No

---

## [Author Response · Author response to Decision Letter 2]

26 Jun 2025

Reviewer #5: The manuscript “Long-term spatial dynamics of jaguars in a high-density population” brings us an excellent analysis of the socio-spatial dynamics of a jaguar population in Belize. I consider the manuscript to be very well written and to have a good statistical approach to the objectives described. After reading the article and the comments that had already been made by three other reviewers in rounds prior to my evaluation, my only suggestions would be these:

1) reduce the introduction by 10-20% as it is too long

We reduced the introduction by ~ 15% and we think it has improved considerably

2) In this paper, the authors could have considered doing spatiotemporal analysis, not just separating spatial and temporal analysis. Please see, for example, the papers of Karanth et al. (2017), Pereira et al. (2024) and Duarte et al. (2025) to get an idea of a possible spatiotemporal approach that considers both the sites of the cameras and the time at which the photo or video was taken, which would give us an idea of whether there is a temporal exclusion between males and females at each of the sampled site. I understand that this would not be the objective of the present study, and the authors can consider this in a future analysis. However, perhaps the authors could mention this in the discussion, considering it as a future approach that could shed more light on socio-spatial dynamics issues.

We read and included the suggested references inserting a statement at the end of the manuscript, now reading:

Lines 615-621. We recognise the recent development of robust methods to assess spatio-temporal interactions between species using an occupancy framework within a single analysis (e.g. (64–66), and the potential to apply it to studies of interactions between individuals. However, in a study of carnivore sympatry, Karanth et al. (65) noted that the method may be biased with low sample sizes. As such, it may be unsuitable for assessing interactions between individuals unless there are sufficient detections of every individual (across years).

References

Duarte HO, Rosalino LM, Toledo JJ, Hilário RR, Carvalho WD. 2024. Spatiotemporal interactions between jaguars (Panthera onca) and their potential prey in Amazonian islands. Ecological Research 40(2):217-227. https://doi.org/10.1111/1440-1703.12522

Karanth KU, Srivathsa A, Vasudev D, Puri M, Parameshwaran R, Kumar NS. 2017. Spatio-temporal interactions facilitate large carnivore sympatry across a resource gradient. Proceedings of the Royal Society B: Biological Sciences 284(1848):20161860. https://doi.org/10.1098/rspb.2016.1860

Pereira R, Matias G, Santos-Reis M, Rosalino LM. 2024. Influence of habitat edges on spatial and spatio-temporal occurrence patterns of mesocarnivores in landscapes dominated by Eucalyptus plantations. Forest Ecology and Management, 572, 122257. https://doi.org/10.1016/j.foreco.2024.122257

Many thanks for all your thoughts and help.

---

## [Editor Report · Decision Letter 2]

26 Aug 2025

Long-term spatial dynamics of jaguars in a high-density population

PONE-D-24-29955R2

Dear Dr. Bart,

We’re pleased to inform you that your manuscript has been judged scientifically suitable for publication and will be formally accepted for publication once it meets all outstanding technical requirements.

Kind regards,

Randeep Singh

Academic Editor

PLOS ONE

Best wishes to authors have done good work.
---

## [Editor Report · Acceptance letter]

PONE-D-24-29955R2

PLOS ONE

Dear Dr. Harmsen,

I'm pleased to inform you that your manuscript has been deemed suitable for publication in PLOS ONE. Congratulations! Your manuscript is now being handed over to our production team.

Kind regards,

on behalf of

Dr. Randeep Singh

Academic Editor

PLOS ONE